# Assessing Deep Learning Methodologies for Automatic Segmentation of the Velopharyngeal Mechanism

**Jiebei Liu**[1]                                    MCU2XN@VIRGINIA.EDU
**Don Brown**[1,2]                                   DEB@VIRGINIA.EDU
**Stephen Baek**[2]                                  BAEK@VIRGINIA.EDU
**Kazlin Mason**[3]                          KAZLIN.MASON@VIRGINIA.EDU

[1] *Department of Systems and Information Engineering, University of Virginia, Charlottesville, VA, US.*

[2] *Data Science Institute, University of Virginia, Charlottesville, VA, US.*

[3] *Department of Human Services, University of Virginia, Charlottesville, VA, US.*

**Editors:** Under Review for MIDL 2023

## Abstract

Velopharyngeal dysfunction (VPD) results in speech, resonance, and swallowing difficulties due to inadequate separation of oral and nasal cavities by the velopharyngeal musculature. Diagnosing and treating VPD often involve multidisciplinary evaluation and specialized imaging techniques like videofluoroscopy or nasendoscopy. However, recent MRI applications have enabled non-invasive visualization of the vocal tract and VP mechanism, providing insights into the shape, size, movement, and position. In order to obtain this data, however, manual techniques are necessary, and analyses of 3D MRI data are time-consuming and not yet clinically feasible. This article aims to explore the feasibility of 3D medical image deep learning methods for segmenting soft palate, levator muscle, pharyngeal wall, and adenoids in the velopharyngeal region, overcoming current limitations and contributing to future clinical translation of this assessment methodology.

**Keywords:** Semantic Segmentation, Velopharyngeal Dysfunction, Velopharyngeal Imaging

## 1. Introduction

The velopharyngeal mechanism is a crucial and complex structure responsible for separating the oral and nasal cavities during speech and swallowing, and its dysfunction (VPD) can be caused by various factors such as, cleft palate and other craniofacial conditions resulting in insufficient palatal tissue, aberrant insertion of the levator veli palatini (LVP) muscle, or a congenitally deep nasopharynx, among others(Woo, 2012). Cleft palate or submucous cleft palate is one of the leading causes of VPD, affecting around 1.5 per 1000 live births(Allam et al., 2014). Even after surgical intervention to repair a cleft, VPD can still persist in approximately 20-30% of patients (Witt et al., 1998; Sullivan et al., 2011) and impair their speech and resonance function.

Measuring the size, shape, and position of the structures involved in achieving velopharyngeal closure is challenging due to the complexity of this region, particularly given the limitation of standard imaging methods such as nasendoscopy and videofluoroscopy. These specialized imaging techniques can be invasive, cause discomfort, and expose children to ionizing radiation. However, accurate assessment is essential for diagnosing and treating VPD. Therefore, non-invasive methods that accurately evaluate velopharyngeal function

are needed in clinical practice. MRI is a versatile and non-invasive imaging technique increasingly used in speech assessment for patients with VPD(Mason and Perry, 2017; Mason, 2022) and has been suggested as a potential tool. However, utilization of MRI data, particularly 3D data analysis, is time-consuming and not yet clinically feasible.

Recently, deep learning-based methods have been developed to segment the entire vocal tract and articulators in MRIs (Ruthven et al., 2021; Erattakulangara and Lingala, 2020). However, these methods mainly focus on 2D MRI and have not been trained to address the underlying velopharyngeal musculature, such as the LVP muscle, velopharyngeal structures, and nasopharyngeal airway.

This pilot study aims to explore the feasibility of applying 3D medical image deep learning methods to segment key velopharyngeal structures and muscles. The proposed method may provide non-invasive anatomical image support to improve the assessment of the velopharyngeal anatomy, reducing the need for invasive assessment procedures and improving patient comfort, especially for children.

## 2. Dataset and Method

We conducted our pilot experiment using 50 T1-weighted whole-head 3D MRI scans in children with normal anatomy collected from the University of Virginia Health System. Segmented annotations for six specific velopharyngeal structures were completed: Adenoids, Lateral Pharyngeal Wall (LPW), Levator Veli Palatini (LVP), Posterior Pharyngeal Wall (PPW), Pterygoid Raphe (PR), and Soft Palate. Figure 1($a$) displays a 3D representation of the annotations. The models were trained using a 32GB NVIDIA V100 GPU for 200 epochs, utilizing DiceCELoss as the loss function. 3D Unet(Çiçek et al., 2016), nnU-Net(Isensee et al., 2020), Swin UNETR(Hatamizadeh et al., 2022), and 3DUX-Net(Lee et al., 2023) models were assessed for accuracy of automated segmentation tasks. All experiments were conducted using five-fold cross-validation schemes with a ratio of 80:20.

## 3. Results

Table 1: Dice score comparison

| Model | Adenoids | LPW | LVP | PPW | PR | Soft Palate | Avg |
|---|---|---|---|---|---|---|---|
| 3DU-Net | 0.6718 | 0.5815 | 0.3970 | 0.6181 | 0.3843 | 0.7071 | 0.5600 |
| nn-UNet | 0.7589 | 0.6374 | 0.5152 | 0.7135 | 0.5025 | 0.8253 | 0.6588 |
| SwinUNETR | 0.7901 | 0.6424 | 0.5592 | **0.7213** | 0.5205 | **0.8541** | 0.6813 |
| 3DUX-Net | **0.8218** | **0.6482** | **0.5638** | 0.7194 | **0.5261** | 0.8344 | **0.6856** |

Table 1 offers a dice score comparative of state-of-the-art(SOTA) transformer and convolutional neural network (ConvNet) models in the context of medical image segmentation for volumetric settings, including 3D Unet, nnU-Net, Swin UNETR, and 3DUX-Net. Figure 1($b$) shows the qualitative representations of tissue segmentation, arranged from top to bottom, displaying the middle slice of the coronal, axial, and midsagittal planes. Each structure is delineated by a unique color for clarity: Adenoids (dark blue), LPW (light

blue), LVP (green), PPW (yellow), PR (pink), and Soft Palate (red).

By implementing minor modifications and fine-tuning, these models have yielded promising results when applied to velopharyngeal data. In particular, the segmentation of the soft palate and adenoids has exceeded a performance metric of 0.80, demonstrating their efficacy for this specific application.

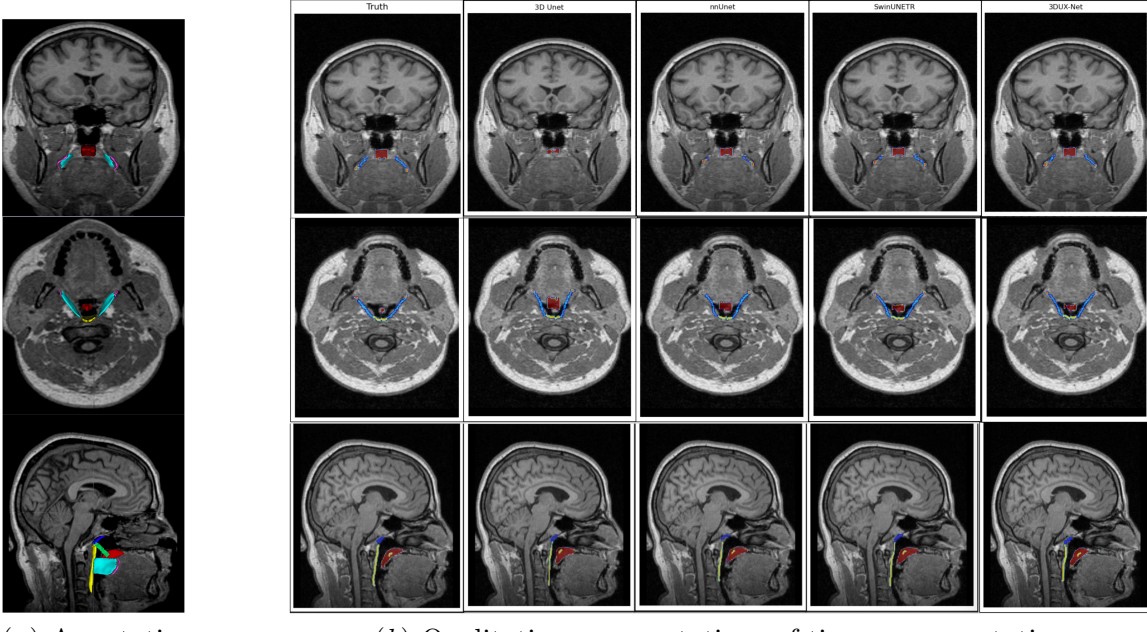

($a$) Annotation        ($b$) Qualitative representations of tissues segmentation

Figure 1: 3D Annotation and Tissues Segmentation

## 4. Conclusions & Outlook

This study used SOTA transformer and ConvNet deep learning models to segment velopharyngeal tissues. Our findings demonstrate that 3D segmentation models have the potential to aid in the anatomical analysis of velopharyngeal anatomy. Using a limited dataset of only 50 T1-weighted MRIs, the existing models achieved promising results, particularly for automatic segmentations of the soft palate and adenoids. With additional data sources, we expect model parameters to improve and result in high accuracy for automated segmentation of the velopharyngeal region.

## Acknowledgments

This work was supported in part by the National Center for Advancing Translational Sciences of the National Institutes of Health under Award Numbers KL2TR003016 (Mason). The content is solely the responsibility of the authors and does not necessarily represent the official views of the National Institutes of Health.

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
