# OpenReview forum: "Assessing Deep Learning Methodologies for Automatic Segmentation of the Velopharyngeal Mechanism"
_MIDL.io/2023/Short_Paper_Track — MIDL 2023 Short paper track Poster_

### Official Review · Reviewer_8iRK · 2023-04-10
**Straightforward but well-performed & informative**

**Rating:** 7
**Confidence:** 5

**Review:**

The authors present automated segmentation results on a new dataset of brain MRI scans with manual labels for velopharyngeal structures, comparing a set of modern segmentation architectures. The work is straightforward but the paper is well-written, the experiments are well-performed, the comparison across architectures is informative, and the domain is novel - meaning, that it would be great for the neuroimaging community if the data were released; I urge the authors to do so.

---

### Official Review · Reviewer_HoGe · 2023-04-17
**I find this work fits better in a more oriented clinical conference**

**Rating:** 4
**Confidence:** 5

**Review:**

This paper attempts to evaluate four 3D state-of-the-art segmentation models for the task of segmenting the entire velopharyngeal mechanism.

Strengths:

- This study may have an impact on the clinical community working on this problem.
- The paper is well written and easy to follow.

Weaknesses:

- Despite being a 3-pager abstract, as this work involves an evaluation of existing methods, I would have expected a bit more depth in the evaluation of the results.
- Authors use the value 0.8 of DSC as a threshold to motivate the effectiveness of different approaches. Is there any clinical evidence that points towards that value? I believe that depending on the application/problem, agreement between expert annotators can be even less.
- Overall, I found that the observations from this work may be of more interest for the clinical community on this topic rather than the standard/average MIDL audience.